# Identifying mRNAs Residing in Myelinating Oligodendrocyte Processes as a Basis for Understanding Internode Autonomy

**DOI:** 10.3390/life13040945

**Published:** 2023-04-04

**Authors:** Robert Gould, Scott Brady

**Affiliations:** 1Whitman Research Center, Marine Biology Laboratory, Woods Hole, MA 02543, USA; 2Departments of Anatomy and Cell Biology, University of Illinois at Chicago, Chicago, IL 60612, USA; stbrady@uic.edu

**Keywords:** myelination, oligodendrocytes, local protein synthesis, mitochondria, endocytosis, Allen Mouse Brain Atlas, Allen Mouse Spinal Cord Atlas, LPAR1, TRP53INP2, TRAK2, TPPP, SH3GL3, endophilin

## Abstract

In elaborating and maintaining myelin sheaths on multiple axons/segments, oligodendrocytes distribute translation of some proteins, including myelin basic protein (MBP), to sites of myelin sheath assembly, or MSAS. As mRNAs located at these sites are selectively trapped in myelin vesicles during tissue homogenization, we performed a screen to identify some of these mRNAs. To confirm locations, we used real-time quantitative polymerase chain reaction (RT-qPCR), to measure mRNA levels in myelin (M) and ‘non-myelin’ pellet (P) fractions, and found that five (*LPAR1*, *TRP53INP2*, *TRAK2*, *TPPP*, and *SH3GL3*) of thirteen mRNAs were highly enriched in myelin (M/P), suggesting residences in MSAS. Because expression by other cell-types will increase *p*-values, some *MSAS* mRNAs might be missed. To identify non-oligodendrocyte expression, we turned to several on-line resources. Although neurons express *TRP53INP2*, *TRAK2* and *TPPP* mRNAs, these expressions did not invalidate recognitions as *MSAS* mRNAs. However, neuronal expression likely prevented recognition of *KIF1A* and *MAPK8IP1* mRNAs as MSAS residents and ependymal cell expression likely prevented APOD mRNA assignment to MSAS. Complementary in situ hybridization (ISH) is recommended to confirm residences of mRNAs in MSAS. As both proteins and lipids are synthesized in MSAS, understanding myelination should not only include efforts to identify proteins synthesized in MSAS, but also the lipids.

## 1. Introduction

One strategy used to expedite the formation and maintenance of specialized structures in polarized cells is to locate protein (and lipid) syntheses near to where the structures form and are maintained [1]. This strategy requires the transport of specific mRNAs along with translation machinery (and lipid enzymes) to these sites [2,3,4]. Myelinating oligodendrocytes (MOs) synthesize a major compact myelin constituent, MBP, in processes overlying sites of compact myelin formation or MSAS, demonstrating availability of translational machinery.

Because of its prominent role in myelin compaction, *MBP* mRNA transport and local translation in oligodendrocyte processes have been the focus of numerous studies [5,6,7,8,9,10,11] and represent the gold standard for mRNAs enriched at the MSAS. However, little effort has been placed in identifying and characterizing other mRNAs transported to and translated in MSAS. To understand roles that locally synthesized proteins play in myelination, one must identify many of the mRNAs located and translated in MSAS. Complementing screening protocols, ISH could confirm mRNAs that co-localize with *MBP* mRNA in MSAS. Published studies have used ISH to show that alternatively-spliced isoforms of myelin-associated oligodendrocytic basic protein (*MOBP*) [12], highly basic proteins that co-reside with MBP in the mammalian CNS myelin major dense line [13], alternatively-spliced isoforms of microtubule associated protein tau (*MAPT*) [14], carbonic anhydrase 2 (*CAR2*) [15], αB-crystallin (*CRYAB*) [16], and kinesin family member 1A (*KIF1A*) [17].

Based on screens and follow-up Northern blot analyses [18], we identified mRNAs encoding: dynein light intermediate chain 2 (DLIC2), peptidylarginine deiminase 2 (PADI2), SH3 domain containing GRB2 like 3/endophilin-A3 (SH3GL3), trafficking kinesin protein 2 (TRAK2, 63NA65*), tumor-promoter p53-inducible nuclear protein-2 (TRP53INP2, SH3-227*) and high molecular weight isoforms of ferritin heavy chain (FTH1). Based on co-purification with *MBP* mRNA in myelin vesicles, many other mRNAs are likely to be located in the MSAS as well [19,20]. However, protocols are needed to distinguish mRNAs that reside in MSAS from contaminants, residing in other intracellular locations.

To address this challenge, we used RT-qPCR (real-time-quantitative polymerase chain reaction) to determine enrichment levels of putative *MSAS* mRNAs identified in our screens (Table 1). *LPAR1*, *TRP53INP2*, *TRAK2*, *TPPP* and *SH3GL3* mRNAs were enriched in myelin at levels approaching levels of *MBP* mRNA. In contrast, levels of *GSN*, *APOD*, *MAPK8IP1* and *KIF1A* mRNA are enriched at lower levels that were similar to mRNAs located in oligodendrocyte somata. Levels of *ENPP2* and *NDRG1* mRNAs were not enriched in myelin and levels of *PADI2* and *DYNC1LI2* mRNAs were too low to determine enrichment. For those mRNAs expressed by other neural cells, enrichments will be lowered, hampering their recognition as MSAS residents. We examined several resources to determine which neural cell types express putative *MSAS* mRNAs, including a widely-cited mouse brain transcriptome, RNA seq database (MBRS, http://www.brainrnaseq.org, accessed on 30 March 2023), Allen Mouse Brain, AMBA (https://mouse.brain-map.org, accessed on 30 March 2023) and Mouse Spinal Cord, AMSCA (http://mousespinal.brain-map.org, accessed on 30 March 2023) Atlases, as well as non-quantitative single oligodendrocyte transcriptome, soRS (http://linnarssonlab.org/oligodendrocytes/, accessed on 30 March 2023) and single cell mouse cortex transcriptome, scRS (http://linnarssonlab.org/cortex/, accessed on 30 March 2023) databases.

In summary, these studies validate the use of myelin subfractions as starting material to identify mRNAs residing at locations where translation products surely will play important and unappreciated roles in CNS myelination. We discuss ways in which this approach could be improved and expanded (see Section 4), providing a template for future studies aimed at determining how local protein synthesis in MSAS participates in internodal myelination. We suggest that efforts to understand internodal myelination will benefit from efforts to characterize lipid synthesis occurring in MSAS.

## 2. Materials and Methods

### 2.1. Animals

CD rats 2-months old, both sexes (Charles River Breeding Laboratory, Wilmington, MA, USA), were used for all studies. Handling and killing of the rats followed protocols that received prior approval from the University of Illinois at Chicago’s Institute’s Animal Welfare Committee. Briefly, the animals were placed in containers containing dry ice and asphyxiated with CO_2_ vapors.

### 2.2. Subcellular Fractionation

The subcellular fractionation method was carried out as described in a previous publication [18]. Briefly, freshly dissected cerebral hemispheres were homogenized in 12.5 mL of ice-cold 0.85 M buffered (10 mm HEPES, pH 7.4 and 3 mM dithiothreitol) sucrose in a 15-mL Dounce Homogenizer. RNase inhibitor (30 u, Five Prime Three Prime, Boulder, CO, USA) was included to prevent RNase degradation. Following homogenization, the samples were centrifuged at 1000 rpm for 10 min to pellet nuclei and cell debris. The low-speed supernatants were removed and added to two 12 mL ultracentrifuge tubes. They were overlayered with a cushion of 6 mL of 0.32 M buffered sucrose and centrifuged in an SW28 swinging bucket rotor at 100,000× *g* for 3.5 h at 4 °C. Myelin vesicles accumulated at the 0.32 M/0.85 M sucrose were collected, dispersed in an equal volume of 10 mM MgCl_2_ and pelleted (12,500 rpm for 20 min. The pellets were also dispersed in 10 mM MgCl_2_. Suspended myelin (M) and pellet (P) fractions were used to prepare total RNA with TRI reagent (Molecular Research Center, Cincinnati, OH, USA) according to protocol from the manufacturer. The purity of the RNA was demonstrated by showing discreet 18S and 28S ribosomal RNA bands following gel electrophoresis.

### 2.3. RT-qPCR Analysis

Total RNAs (0.5 to 1 μg) from myelin and pellet subcellular fractions were used as template to prepare cDNAs. Reactions were performed with SuperScript III (Gibco BRL, Rockville, MD, USA) and oligo dT according to the manufacturer’s protocol. Semi-quantitative polymerase chain reaction q(PCR) was carried out in an iCycler Detection System (Bio-Rad) according to manufacturer’s instruction manual. Ninety-six well plates were used with a SuperMix PCR kit (Bio-Rad, Hercules, CA, USA). Routinely eight primer pairs, always including a GAPDH (300 nM forward and reverse primers, Appendix A) control, and four samples (cell culture or subcellular fractionation RNA, run in triplicate) were used for each experiment. Amplification conditions were 95 °C for 5 min followed by 40 cycles at 94 °C, 60 °C and 72 °C each for 20 s, and a final extension at 72 °C for 7 min. Fluorescence emission was monitored and the threshold (CT) values recorded and averaged. At least three different samples were prepared from subcellular fraction and cell culture experiments.

### 2.4. Analysis of MBRS Data

Data were obtained online from an Excel spread (Appendix A) for each mRNA listed. We selected mRNAs that were specific for myelinating oligodendrocytes (MOs), newly formed oligodendrocytes (NFOs), oligodendrocyte progenitor cells (OPCs), neurons, astrocytes, microglia and ependymal cells, and then calculated percentages of total expressions by each neural cell type and some combinations as listed as well as NFO/OPC and MO/NFO ratios.

### 2.5. Analyses of Expression Patterns for mRNAs in AMBA and AMSCA Images

AMBA and AMSCA archive ISH images for the majority of genes/mRNAs in the mouse genome. Here, we use these databases to determine and compare expression patterns among *MSAS* mRNAs (Section 3.4). In order to make these comparisons, we used mRNAs that were specific for each neural cell type as identified using MBRS data (Appendix A), examining expression patterns in similar regions of sagittal sections from P56 mouse hippocampus, cerebellum and spinal cord and from P4 mouse spinal cord for each mRNA (Section 3.3). Anatomical regions are labeled in the uppermost rows for each tissue (ISH images). Images of representative mRNAs for each cell type were used to identify similarities and differences in cell-type specific expression patterns (Section 3.3). Positive expressions by myelinating oligodendrocytes (O+), newly-formed oligodendrocytes (O+), oligodendrocyte progenitor cells (O+), neurons (N+), astrocytes (A+), microglia (M+), ependymal cells (E+) and spinal cord meninges (M+) were labeled over regions showing mRNA expression. Regions lacking cell-type specific expression were similarly labeled (O−, O−, O−, N−, A−, M−, E− and M−) as were regions in which we could not decide whether expression did or did not occur (O**±**, O**±**, O**±**, N**±**, A**±**, M**±**, E**±**
and M**±**). We included summaries showing the cell types that expressed the mRNAs in leftmost panels, with the top row recording expected cell expression and lower rows showing expression by additional cell types. These summary panels were boxed in colors identifying the cell types that expressed the mRNAs.

Although difficult to detect mRNA expression in MSAS, i.e., in white and gray matter areas around oligodendrocyte somata with the exception of highly-expressed *MBP* mRNA (Section 3.3), we felt it was worth examining expression more closely in high-magnification ISH images (Section 3.5). We contrasted *MBP* mRNA (MSAS) expression with *CNP* mRNA (somata) and with *MOBP* mRNA (some isoforms reside in MSAS) and then looked for MSAS expression of the *MSAS* mRNAs which we had showed located to MSAS (Section 3.1). Clearly, whereas *MBP* mRNA expression is easy to see in P56 mouse corpus callosum and fornix, cerebellar white matter and spinal cord, it is not possible to identify expression of any *MSAS* mRNAs in high-magnification images.

### 2.6. Statistical Analysis

Statistically significant enrichment of mRNAs in myelin vesicles (Section 3.1) was evaluated using a one sample t and Wilcoxon test in GraphPad Prism 9 for MacOS.

## 3. Results

The overall goal of this study is to identify and characterize mRNAs which, based on co-fractionation with *MBP* and *MOBP* mRNAs in myelin vesicles, reside in MSAS. Locating mRNAs to MSAS enables translations at sites where internodal myelin sheaths form and function, and where axon-oligodendrocyte signaling can influence translation.

Previously, we identified putative *MSAS* mRNAs using suppression subtractive hybridization (SSH) on mRNAs prepared from starting homogenates and from myelin isolated by subcellular fractionation of 2–3-month rat cerebral hemispheres. In these studies, we used Northern blot to show that *PADI2*, *SH3GL3*, *TRAK2*, *TRP53INP2*, *KIF1A*, *DYNC1LI2* and high molecular weight isoforms of *FTH1* mRNA reside in MSAS, based on enrichments in myelin fractions over starting material [17,18]. Here, we extend these findings using RT-qPCR to determine enrichments in myelin over a ‘non-myelin’ pellet subfraction.

Whereas known *MSAS* (*MBP* and *MOBP*) mRNAs are selectively expressed by differentiated oligodendrocytes (DOs) [42,43], other putative *MSAS* mRNAs may be expressed by other neural cell types. If so, methods which we use to identify *MSAS* mRNAs might be compromised. To find out which putative *MSAS* mRNAs are expressed by non-oligodendrocyte lineage cell-types, we determined cell-types expressing these mRNAs using a mouse brain transcriptome, RNA seq (MBRS) database (Section 3.2), and in Allen Mouse Brain Atlas (AMBA) and Mouse Spinal Cord Atlas (AMSCA) databases (Section 3.4). We also examined expressions using non-quantitative single cell transcriptome RNA seq (scRS) and single oligodendrocyte cell transcriptome RNA seq (soRS) databases.

### 3.1. Enrichments of MSAS mRNAs in Rat Brain Myelin Based on RT-qPCR

As in previous studies [18,44], we used cerebral hemispheres from 2–3-month-old rats (both sexes) as starting material, and myelin and pellet subfractions from tissues homogenized in hypertonic (0.85 M) sucrose. We used this condition because enrichments of *MBP* and *MOBP* mRNAs in myelin over starting homogenate were slightly better than subfractions prepared from tissues homogenized in isosmotic sucrose [44,45]. Using mRNA specific primers (Appendix A) for RT-qPCR, we measured abundances of mRNAs resident in MSAS, oligodendrocyte somata, neurons and astrocytes and compared our results with ones obtained in a study that used transcriptome, RNA seq to measure levels of mRNAs in P72 mouse brain myelin (Table 2).

We compare results from our SSH study with results from an mmRS study that identified mRNAs present in P72 mouse brain myelin [20]. Differences may arise from: (1) tissues, rat cerebral hemispheres (SSH) versus mouse brains (mmRS); (2) homogenization solutions (hypertonic (SSH) versus isosmotic (mmRS)); (3) analyses (SSH versus mmRS); and (4) post-myelin purification, where hypo-osmotic shock was used in the mmRS study. As results are in different units, µg/µL (SSH) and normalized read count (mmRS), data sets were normalized to *MBP* mRNA (%*MBP*, Table 2). To aid dataset comparisons we include SSH/mmRS ratios of %*MBP* mRNA.

In both studies, *MBP* mRNA was by far the most abundant mRNA in isolated myelin. *MOBP* and *PLP1* mRNAs were more abundant, ~2.5% (SSH) and ~7% (mmRS) the levels of *MBP* mRNA, respectively, than other mRNAs. Although selectively expressed in oligodendrocyte somata [12,46], levels of *CNP* and *MOBP-81B* (SSH) mRNAs in myelin were not much greater than levels of neuronal (*NEFL*) and astrocyte (*GFAP*) mRNAs (>0.1% the level of *MBP* mRNA in myelin fractions of both SSH and mmRS), suggesting that mRNAs abundant in neurons, astrocytes and their processes in the vicinity of compact myelin are trapped in myelin during homogenization. In both studies, *LPAR*1 and *NDRG1* mRNAs were more abundant (0.2–0.7% the level of *MBP* mRNA) than other *MSAS* mRNAs. These results indicate that differences in tissues and in protocols in the two studies have limited influence on the populations of mRNAs trapped in the myelin vesicles and purified by floatation on 0.85 M sucrose.

In previous studies, we used semi-quantitative Northern blots to show that some of the mRNAs identified in our screen were enriched in myelin (Table 1). Unfortunately, the mmRS study relied solely on abundance data and failed to include normalization. Here, we use RT-qPCR, normalizing myelin (M) RNA values to ‘non-myelin’ pellet (P) RNA values obtained in the same fractionations (M/P ratios, Figure 1). Unlike *MBP* and *MOBP* mRNAs, which are specifically expressed by differentiated oligodendrocytes, some mRNAs identified in our screens are expressed by other neural cells, i.e., neurons, astrocytes, microglia and/or ependymal cells. Consequently, mRNAs located in these cells will add to P (pellet)-values and, consequently, reduce enrichments (M/P).

Unlike abundance data (Table 2), M/P ratios distinguish MSAS-residing *MBP* and *MOBP-81A* mRNAs (37-fold enriched) from oligodendrocyte somata mRNAs (*PLP1*, *MOBP-81B* and *CNP* mRNAs, 5-fold enriched), and from neuron (*NEFL*, not enriched) and astrocyte (*GFAP*, not enriched) mRNAs (Figure 1).

**Figure 1 life-13-00945-f001:**
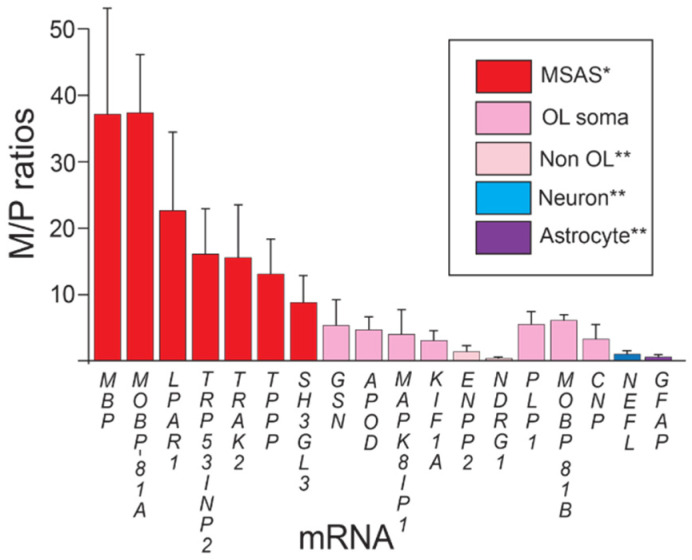
Enrichments of mRNAs in myelin vesicles. * Statistically > oligodendrocyte somata mRNA (*p* > 0.01). ** Statistically <oligodendrocyte somata mRNA (*p* < 0.01). Enrichments (M/P ratios ± standard deviations) of mRNAs in myelin vesicles prepared by subcellular fractionation (Methods). Plots show mRNAs grouped according to enrichments: MSAS, oligodendrocyte somata, non-oligodendrocyte, neuron and astrocyte mRNAs.

Using 5-fold enrichment levels of oligodendrocyte somata mRNA as baseline, we find that, like *MBP* and *MOBP-81A* mRNAs, *LPAR1*, *TRP53INP2*, *TRAK2*, *TPPP* and *SH3GL3* mRNAs are significantly enriched in myelin over oligodendrocyte somata mRNAs, supporting their residences in MSAS (Figure 1). One reason why enrichments of these *MSAS* mRNAs are lower than *MBP* and *MOBP-81A* mRNAs might be because lower portions reside in MSAS, i.e., higher portions are retained in oligodendrocyte somata. Actually, as seen below, enrichments of *TRP53INP2*, *TRAK2* and *TPPP* mRNAs are likely higher as they are not only expressed by oligodendrocytes, but also by neurons (Section 3.2 and Section 3.4). Expression by neurons [47,48], also Section 3.2 and Section 3.4, likely brings the enrichment value of *KIF1A* mRNA below the level demonstrating residence in MSAS (Figure 1). Similarly, because *APOD* mRNA is expressed by ependymal cells (Section 3.2 and Section 3.4), its enrichment may be lowered to levels below those needed to demonstrate residence in MSAS. Using ISH, we proved that *KIF1A* mRNA resides in MSAS [17].

Although myelinating oligodendrocytes express *NDRG1* and *ENPP2* mRNAs [23,29], and both mRNAs are trapped in myelin during tissue homogenization (Table 2), neither mRNA was enriched in myelin, not even to levels of oligodendrocyte somata mRNAs (Figure 1). As other neural cell-types do not express these mRNAs (Section 3.2 and Section 3.4), the only explanation we have for their lack of enrichment is that the hypertonicity of the homogenization solution used in this study selectively blocks entrapments of *NDRG1* and *ENPP2* mRNAs in myelin. Although we don’t have the data, we believe that these mRNAs were identified in screens of tissues homogenized in isosmotic sucrose.

### 3.2. Identifying Neural Cell-Types Expressing MSAS mRNAs Using MBRS Data

To determine the cell-types that express putative *MSAS* mRNAs (Table 1), we used MBRS data [49], as this resource contains quantitative expression data for mRNAs of seven neural cell-types acutely isolated from developing mouse brain (Figure 2). We include mRNAs expressed by MOs (*MBP*), NFOs (*MYRF*/*GM98*), OPCs (*PDGFRA*) and neurons (*TUBB3*) mRNAs for comparison (Figure 2), mRNAs also used in assessing expression patterns in AMBA and AMSCA images (Figure 3). MBRS data verify expressions as revealed through differences in MO/NFO ratios for *MBP* mRNA (3.6×) and *MYRF* mRNA (0.7×), through OPC/NFO ratios for *PDGFRA* mRNA (7.5×) and through neuron/(OPC + NFO + MO) ratios for *TUBB3* mRNA (9.2×), results verified with scRS data [50].

We comment that expression levels of *MBP* mRNA and other *MSAS* mRNAs obtained in MBRS data are from acutely isolated oligodendrocyte somata lacking the processes where *MBP* mRNA and other *MSAS* mRNAs reside, i.e., values are likely underestimates. Nevertheless, NFOs and MOs highly and somewhat selectively express *MBP* and *MSAS* mRNAs (Figure 2). Expression levels in MOs vary more than 60-fold, from *GSN* mRNA at ~1500 Fragments per Kilobase of transcript per Million mapped reads (FPKM) to *PADI2* mRNA at 25 FPKM. The *MSAS* mRNAs with lowest expressions, *DYNC1LI2* and *PADI2* mRNAs, are the ones present at too low abundances to quantify by RT-qPCR (Section 3.1).

From MBRS data, differentiated oligodendrocytes selectively express mRNAs (*LPAR1*, *TRP53INP2*, *TRAK2*, *TPPP* and *SH3GL3* mRNAs, >82% of total, Figure 2) present in MSAS (Figure 1) and others (*GSN*, *ENPP2*, *NDRG1*, *PADI2* and *CRYAB* mRNAs, >84% of total, Figure 2). Neurons express *MAPK8IP1*, *KIF1A* and *DYNC1LI2* mRNAs, astrocytes express *KIF1A*, *DYNC1LI2* and *CAR2* mRNAs and ependymal cells express *APOD* mRNA. ScRS data confirm neuronal expressions of *MAPK8IP1*, *KIF1A*, and *DYNC1LI2* mRNAs as well as *TRP53INP2*, *TRAK2*, *TPPP* and *NDRG1* mRNAs (*, Figure 2), findings indicating enrichments of these mRNAs are lowered by neuron contributions. Neuronal expressions were confirmed in AMBA and AMSCA images (Section 3.4). ScRS data also identified expression of *PADI2* mRNA by astrocytes, though not expressions of *KIF1A*, *DYNC1LI2* and *CAR2* mRNAs by astrocytes, nor expression of *APOD* mRNA by ependymal cells, indicating it is best to examine as many resources as possible.

As illustrated from MO/NFO ratios, expressions of *TPPP*, *ENPP2*, *KIF1A*, *DYNC1LI2*, *PADI2*, *LPAR1, TRAK2* and *SH3GL3* mRNAs (0.8–1.7) occur earlier than expressions of *TRP53INP2*, *GSN*, *APOD, CAR2*, *CRYAB* and *NDRG1* mRNAs (1.9–4.8). These differences indicate that the different mRNAs participate in different stages of myelination and support a need to identify *MSAS* mRNAs from tissues at relatively early stages of myelination (Section 4.1).

Expression levels in NFO and MO somata from P7–P12 mouse brain used for MBRS may differ from levels in the myelinating oligodendrocyte processes from older (60–90-day) rat cerebral hemispheres used for the SSH screens and P18 and P72 mouse brains used for the mmRS study. As AMBA and AMSCA images are from P56 animals, they may better reflect expression from tissues used for SSH screens.

To correctly identify cells that express different *MSAS* mRNAs, we examined and compared expression patterns of mRNAs selectively expressed by different neural cell-types identified using AMBA and AMSCA data (next section).

### 3.3. Recognizing Similarities and Differences among Neural Cell-Type Specific mRNA Expression Patterns in AMBA and AMSCA Images

To understand cell types expressing *MSAS* mRNAs using AMBA and AMSCA images we examined expression patterns of several MO-, NFO-, OPC-, neuron-, astrocyte-, microglia- and ependymal cell-mRNAs (Appendix A). The *MO* mRNAs selected (*MBP*, *MOBP* and *MAL*) were expressed at higher levels than mRNAs specific to other cell-types. They were expressed at 67–78% of total by MOs and >99% of total by oligodendrocyte lineage (OL) cells. *NFO* mRNAs selected (*GPR17* and *MYRF*), were expressed at 58–64% of total by NFOs and >99% of total by OLs. *OPC* mRNAs (*PDGFRA* and *CSPG4*) were expressed at 76–83% of total by OPCs and >91% of total by OLs. Neuronal mRNAs (*TUBB3* and *STMN2*) were expressed at 87–92% of total by neurons. Astrocyte mRNAs (*AQP4* and *ALDH1L1*) were expressed at 86–92% of total by astrocytes. Microglia (*C1QA*) and ependymal cell (*CLDN5*) mRNAs were expressed at 96% of total by these cells.

Here, we examine ISH and corresponding experimental analysis (EA) images in sagittal sections of P56 mouse hippocampus and cerebellum (AMBA) and P56 and P4 spinal cord hemisections (AMSCA). Links to datasets are given in Appendix A. In these tables, we show assignments of regions that express the mRNAs provided in AMSCA datasets.

#### 3.3.1. Regions Occupied by Oligodendrocyte Lineage Cells—AMBA and AMSCA Data

We examined AMBA and AMSCA images as complements to MBRS data, as they provide expressions by cells in situ. We believe these resources are underutilized and by careful examination of expression patterns of mRNAs with known cell-type specificity (Figure 3), they provide a framework for identifying all the cell-types that express putative *MSAS* mRNAs (Section 3.4).

**Figure 3 life-13-00945-f003:**
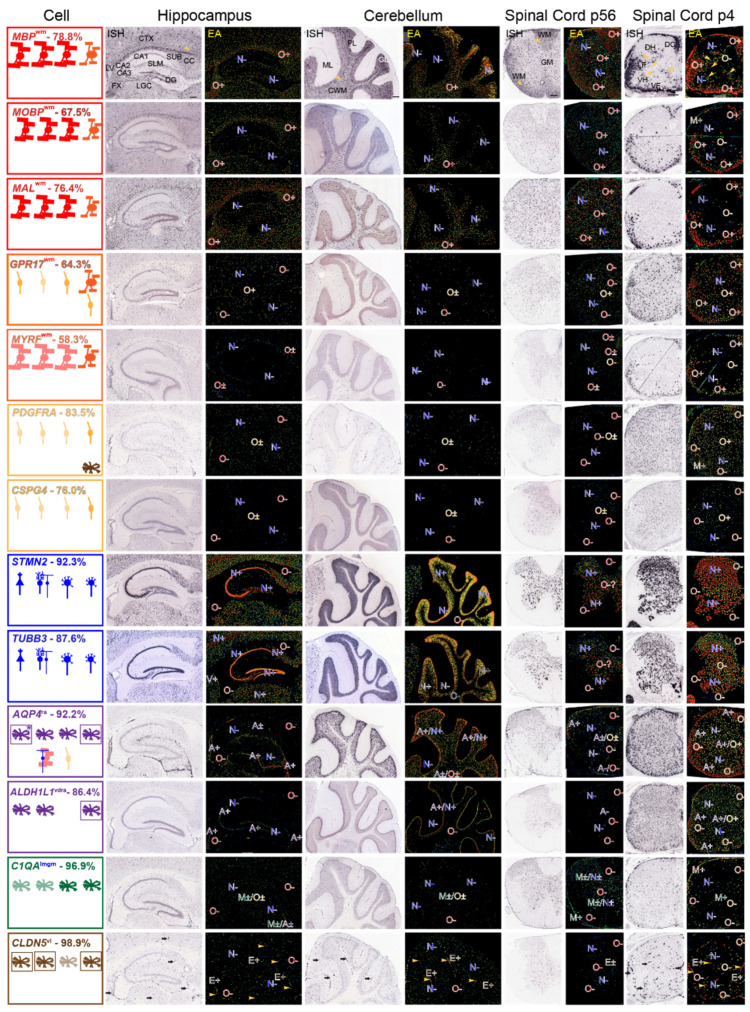
Expression patterns of mRNAs representing each neural cell-type—AMBA and AMSCA images. ISH and EA images of mRNAs representing each cell-type in sections of P56 mouse hippocampus and surrounding tissue, cerebellum and spinal cord and of P4 spinal cord. Regional assignments (AMSCA, Appendix A) are listed as superscripts following the gene name. Summaries show cell-types expression of mRNAs in leftmost panels, arranged from left-to-right (hippocampus, cerebellum, P56 and P4 spinal cord). Expected expression is in the uppermost row and additional cells expressing the mRNA are shown in the middle and lower rows. Leftmost figures are boxed with colors which identify the prominent cell-types that express the mRNA. Where astrocyte and ependymal cell mRNA expressions are clearest, cells in leftmost panels are boxed. In EA images throughout, expressions are marked as indicated in Materials and Methods (Section 2.4). Abbreviations: in ISH images of *MBP* mRNA: hippocampus—and surrounding corpus callosum (CC), fornix (FX), cortex (CTX), lateral geniculate complex (LGC), subiculum (SUB), stratum lacunosum moleculare (SLM), CA1, CA2, CA3, dentate gyrus (DG), lateral ventricle (LV); cerebellum—molecular layer (ML), Purkinje cell layer (PL), granule cell layer (GL), cerebellar white matter (CWM); P56 spinal cord—white (WM), gray (GM) matter; P4 spinal cord—dorsal column (DC), lateral (LF) and ventral (VF) funiculi, dorsal (DH), ventral (VH) horns. Scale bars listed for *MBP* mRNA are 200 μm.

As OPCs, NFOs and MOs are three different stages of oligodendrocyte development, and we are using AMBA and AMSCA images to both link *MSAS* mRNAs to oligodendrocyte lineage cells and to other neural cells, understanding where the cells expressing cell-specific mRNAs locate in tissue greatly enhances our understanding of how the distributions of cells expressing *MSAS* mRNAs will influence enrichments. With AMSCA data from P4 spinal cord, they will also provide information as to when in development, they participate in myelination.

We examined distribution patterns of *MBP*, *MOBP* and *MAL* mRNAs to determine where *MO* mRNAs reside in and around the hippocampus, in the cerebellum and in spinal cord of P56 mouse and in spinal cord of P4 mouse. Protein products of all three mRNAs reside in compact myelin [51]. Based on MBRS data, all are specifically expressed by MOs (77.8%, 67.5% and 76.4%, respectively, of total and >99% of total as *OL* mRNAs, Appendix A). All locate to white matter in AMSCA images (Figure 3, Appendix A). All are widely expressed throughout mouse brain and spinal cord (Figure 3). Compared with relatively low expressions in the hippocampus per se, these mRNAs are highly expressed by MOs concentrated in the corpus callosum and fornix (O+, Figure 3) and spread throughout the cortex and lateral geniculate complex. Expression by a few oligodendrocytes in the hippocampus is readily distinguishable from expressions of *TUBB3* and *MAP2* mRNAs by neurons located in CA1–3 and dentate gyrus (N+, Figure 3). In cerebellum, *MBP, MOBP* and *MAL* mRNA-expressing MOs concentrate in white matter (O+, Figure 3) and spread into the granule cell layer. Neurons in molecular, granule and Purkinje cell layers do not express these mRNAs (N−, Figure 3). In P56 spinal cord, *MO* mRNA-expressing cells disperse throughout white and gray matter (O+, Figure 3). Only *MBP* mRNA shows higher white matter expression, which likely relates to expression in MSAS (see below). Neurons, in gray matter, including large-profile motoneurons in the ventral horn, do not express these mRNAs (N−, Figure 3). Summaries (leftmost panels) shows that *MO* mRNA-expressing cells reside in P56 corpus callosum and fornix, and in cerebellar white matter and spinal cord white and gray matter (
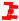
, Figure 3). In P4 spinal cord, *MO* mRNA-expressing NFOs are largely restricted to white-matter-rich ventral and lateral funiculi and dorsal columns (O+, Figure 3), as depicted in the leftmost panels (
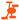
, Figure 3). Neither OPCs nor neurons in P4 spinal cord gray matter express these mRNAs, (N−, O−, Figure 3), as evidenced by comparisons with expression patterns of OPC and neuronal mRNAs (see below). For *MBP*, though not for *MOBP* or *MAL*, mRNAs, surrounding the few gray matter MO somata (red spots depicting high expression in the cells, EA images, Figure 3) are oligodendrocyte-processes/MSAS expressing *MBP* mRNA (green and yellow strands, EA images, arrowheads, Figure 3).

We chose *GPR17* and *MYRF* mRNAs, both selectively expressed by NFOs (64.3% and 58.3% of total, respectively and showing MO/NFO ratios < 0.7), to determine regions where NFOs reside. Considering early *GPR17* mRNA expression by OPCs (30.5% of total, Appendix A) and persistent *MYRF* mRNA expression by MOs (38.7% of total, Appendix A) allows one to see overlaps with OPCs and MOs. Both show the same white matter specificity in AMSCA (Figure 3, Appendix A), i.e., NFOs in P4 white matter (ventral and lateral funiculi and dorsal columns) express high levels of both *GPR17* and *MYRF* mRNAs (O+, Figure 3) as depicted in leftmost panel (
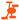
). OPCs spread throughout P4 spinal cord gray matter express *GPR17* mRNA (O−, Figure 3), whereas MOs in corpus callosum and fornix express *MYRF* mRNA (O+, Figure 3). Brain and spinal cord neurons express neither *GPR17* nor *MYRF* mRNAs (N−, Figure 3).

We chose *PDGFRA* and *CSPG4* mRNAs, well-established OPC markers [52,53], to determine where cells expressing *OPC* mRNA reside. Both are selectively expressed by OPCs in the MBRS database (83.5% and 76.0% of total, respectively, Appendix A). Both *PDGFRA* and *CSPG4* mRNAs are selectively expressed by OPCs spread throughout P4 spinal cord white and gray matter (O+, Figure 3, depicted in leftmost panels, 
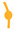
) and by adult OPCs distributed at low density [54,55] throughout P56 brain and spinal cord (O±, Figure 3, in leftmost panels, 
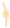
). Neither MOs (O−, Figure 3), NFOs (O−, Figure 3) nor neurons (N−, Figure 3) express *PDGFRA* nor *CSPG4* mRNAs.

#### 3.3.2. Regions Occupied by Non-Oligodendrocyte Lineage Cells—AMBA and AMSCA Data

We chose *STMN2* and *TUBB3* mRNA to identify neuronal mRNA distributions. Both, well-characterized neuronal markers [56,57,58,59], are selectively expressed by neurons (91.9% and 86.4% of total expression, respectively, Appendix A). AMBA and AMSCA images support selective neuronal expressions throughout brain and spinal cord (N+, Figure 3). In hippocampus, both mRNAs locate to neurons concentrated in CA1–3 and in the dentate gyrus (N+, Figure 3), denoted as pyramidal cells in leftmost panels (
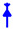
, Figure 3) and to neurons spread throughout the cortex, lateral geniculate complex and subiculum. In cerebellum, Purkinje (
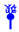
) and granule cell (
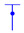
) neurons express both *STMN2* and *TUBB3* mRNAs (N+, Figure 3), though neurons in the molecular layer do not (N−, Figure 3). In P56 and P4 spinal cord, neurons expressing *STMN2* and *TUBB3* mRNAs (N+, Figure 3) are distributed throughout gray matter, including in large profile ventral horn motoneurons (
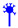
, Figure 3, leftmost panel). Importantly, neither *STMN2* or *TUBB3* mRNAs are expressed in P56 mouse brain and spinal cord white matter where MOs reside (O−, Figure 3), nor in P4 spinal cord white matter (O−, Figure 3) where NFOs reside, allowing clear distinctions between mRNAs expressed by MOs and NFOs, from mRNAs expressed by neurons.

To determine whether astrocytes, microglia and ependymal cells overlap oligodendrocytes and neurons, we examined distributions of well-characterized marker mRNAs. *AQP4* [60] and *ALDH1L1* [49] mRNAs were used to identify regions where astrocytes reside, while *C1QA* [61] and *CLDN5* [62] mRNAs were used to identify regions where microglia and ependymal cells reside. *AQP4* and *ALDH1L1* mRNAs are selectively expressed by acutely isolated astrocytes (91.9 and 86.4% of total expression, respectively, Appendix A). Both were assigned to a category “ra”, representing white matter and radially arrayed white matter (Figure 3, Appendix A). Expression patterns of *AQP4* and *ALDH1L1* mRNAs are different from expression patterns of *MO*-, *NFO*-, *OPC*- and neuronal mRNAs (Figure 3). They were not expressed in MO-rich white (O−, Figure 3) nor in neuron-rich gray (N−, Figure 3) matter. Expression patterns are different for these astrocyte mRNAs, making it difficult to identify astrocyte-specific regions.

The most astrocyte-specific expression occurs in meninges surrounding the fornix, and in the meninges and outer portions of ventral and lateral funiculi in P4 spinal cord (A+, Figure 3). Both *AQP4* and *ALDH1L1* mRNAs are expressed by Bergmann glia in the Purkinje cell layer [63] (A+/N+, Figure 3).

*CIQA* mRNA, a well-characterized microglia marker [61,64] is selectively expressed by microglia in MBRS (96.9% of total expression, Appendix A). Unlike other cell type-specific mRNAs, *CIQA* mRNA is poorly expressed in mouse brain and spinal cord (M-, Figure 3), and consequently expression by microglia is not expected to influence expressions by other cell-type-specific mRNAs. Additionally, it will be difficult to use AMBA and AMSCA images to assign expression to microglia.

*CLDN5* mRNA, a well characterized ependymal cell marker [62,65], is selectively expressed by ependymal cells (98.9% of total, Appendix A) and in regions assigned to vasculature (vl, Appendix A). *CLDN5* mRNA is characterized by distinct expression patterns in brain and spinal cord (E+, Figure 3). Fortunately, *CLDN5* mRNA-expressing cells do not overlap with MOs (O−, Figure 3), NFOs (O−, Figure 3), OPCs (O−, Figure 3) or neurons (N−, Figure 3). *CLDN5* mRNA-expressing ependymal cells form distinct cell clusters (arrowheads, Figure 3) that run along the vasculature and are more abundant in and around the hippocampus, in the cerebellar molecular layer, especially in P4 white and gray matter.

### 3.4. Regions in AMBA and AMSCA Images Where MSAS mRNAs Are Expressed

MOs and/or NFOs express all putative *MSAS* mRNAs (Figure 4). MOs in brain and spinal cord robustly express *GSN*, *APOD*, *ENPP2*, *NDRG1*, *CAR2* and *CRYAB* mRNAs (O+, Figure 4) in accordance with ratings given in AMSCA (Figure 4, Appendix A), and at lower intensity *LPAR1*, *TPPP*, *MAPK8IP1*, *KIF1A* and *PADI2* mRNAs (O**±**, Figure 4). MOs do not express *TRP53INP2*, *TRAK2*, *SH3GL3* or *DYNC1LI2* mRNAs (O−, Figure 4) at detectable levels, suggesting they associate with active myelination. NFOs in P4 spinal cord robustly express *TRAK2*, *TPPP*, *GSN*, *MAPK18P1*, *KIF1A* and *DYNC1LI2* mRNAs (O+, Figure 4) and less intensely, *LPAR1*, *TRP53INP2*, *SH3GL3*, *APOD* and *CAR2* mRNAs (O**±**, Figure 4). NFOs do not express *NDRG1* and *CRYAB* mRNAs (O−, Figure 4), suggesting expressions associate with oligodendrocyte maturation. These temporal patterns are generally similar to expression patterns found with MBRS data (Figure 2). In particular, both datasets show that *TPPP*, *SH3GL3* and *DYNC1LI2* mRNAs are expressed early and that *NDRG1* and *CRYAB* mRNAs are expressed late.

AMBA and AMSCA images highlight high neuronal expressions of *TRP53INP2*, *TRAK2*, *TPPP*, *MAPK8IP1*, *KIF1A* and *DYNC1LI2* mRNAs. It seems reasonable that prominent neuronal expressions of these mRNAs reduce the likelihood of identifying them as residing in MSAS. Nevertheless, *TRP53INP2*, *TRAK2* and *TPPP* mRNAs locate to MSAS by enrichments (Figure 1), indicating levels in MSAS are even higher than indicated by enrichments.

MOs and NFOs selectively express *LPAR1* and *SH3GL3* mRNAs, with the latter in P4 spinal cord only (O+, O+, Figure 4). Among other *MSAS* mRNAs, ependymal cells express *APOD* mRNA (Figure 4, E+), and astrocytes express *CAR2* and *CRYAB* mRNAs (A+, Figure 4). High ependymal cell expression of *APOD* mRNA may have masked its residence in MSAS.

In summary, AMBA and AMSCA images complement and extend MBRS, scRS and soRS transcriptome data in providing temporal information as to when *MSAS* mRNAs are expressed, as well as identifying ones expressed by neurons, astrocytes and/or ependymal cells.

### 3.5. Using AMBA and AMSCA Images as an Attempt to Identify MSAS mRNAs

Differences in distribution of MSAS-associated *MBP* mRNA and non-MSAS-associated *CNP* mRNA in P56 mouse brain caudate putamen have been recorded using high-magnification ISH images [66]. Additionally, similar high-magnification images identify dendritic residences of calcium/calmodulin dependent protein kinase 2 (*CAMK2*), dead end protein homolog 1 (DND1), microtubule associated protein 1A (*MAP1A*), microtubule associated protein 2 (*MAP2*), inositol 1,4,5-triphosphate receptor type 1 (*ITPR1*) and SH3 domain binding kinase 1 (*SBK1*) mRNAs [66]. The success in visualizing these process-associated expressions led us to see if we could visualize *MSAS* mRNAs in myelinating oligodendrocyte processes.

We examined high-magnification images of *MBP* mRNA distribution in the fornix, cerebellar white matter, and in P56 and P4 spinal cord white matter and compared them with images of *MOBP*, *CNP* mRNAs, *LPAR1*, *TRP53INP2*, *TRAK2*, *TPPP* and *SH3GL3* mRNAs (Figure 5).

At low-magnification, *MBP* mRNA in fornix is concentrated over many labeled cells (black spots) spread throughout the tissue. At high magnification, surrounding the labeled soma are gray areas that likely represent expression in MSAS (arrows in first column, second picture from the top, Figure 5). Similar distribution patterns are seen in high-magnification images of *MBP* mRNA in cerebellar, P56 and P4 spinal cord white matter (arrows in first column images, Figure 5). In P4 spinal cord, *MBP* mRNA expression in gray matter includes thin process around a few somata (arrow, Figure 5, also Figure 3).

This notion that the gray areas surrounding highly expressing somata represent expression in MSAS is strengthened by comparing distribution patterns of *MBP* mRNA with those of *CNP* mRNA, an mRNA largely limited to oligodendrocyte somata [46,67]. Both low and high magnification images of fornix, cerebellar white matter and spinal cord (P56 and P4) white matter show prominent expression in punctate oligodendrocyte somata, and little expression in white matter surrounding the somata. Arrowheads in images of fornix and cerebellum show that, compared with *MBP* mRNA expression, the label is not clear in regions surrounding labeled soma (Figure 5). Similar distribution patterns were seen in high-magnifications images of *PLP1* and *CLDN11* mRNAs. It is difficult to tell whether or not faint gray areas of *CNP* mRNA expression around somata are due to presence in MSAS (arrows, Figure 5). Examining similar images of *PLP1*, *MAG* and *MAL* mRNAs in P4 spinal cord, it was impossible to decide whether small amounts of expressed mRNAs located to MSAS or not. Preparing and examining our own high-resolution images (Section 4.4.1) should provide clear understanding as to whether some of these mRNAs reside in MSAS.

Although a majority of *MOBP* mRNA isoforms are located in MSAS in rat tissues [12,45], the portions of *MOBP* mRNA isoforms residing in mouse MSAS has not been studied. We mention this because portions of *MOBP* mRNA residing in MSAS in AMBA and AMSCA images of mouse brain and spinal cord appear far lower than portions of *MBP* mRNA in MSAS (Figure 5). Nevertheless, in P56 fornix and cerebellar white matter, though not in P56 spinal cord white matter, some *MOBP* mRNA expression appears in white matter surrounding labeled somata (arrowheads, Figure 5). In P4 spinal cord, more pronounced *MOBP* mRNA expression appears in both white and gray matter (arrows, Figure 5), though further studies will be needed (Section 4.4.1) to know with certainty.

Expressions of *LPAR1*, *TRP53INP2*, *TRAK2*, *TPPP* and *SH3GL3* mRNAs in fornix, cerebellar and P56 and P4 spinal cord appear to be restricted to oligodendrocyte somata (Figure 5). However, in all instances, overall expression levels are far lower than expression levels of *MBP*, *CNP* and *MOBP* mRNAs, indicating that MSAS-associated expression might have been identified if exposure times were increased. In a few cases, small clusters of expression are seen (arrowheads, Figure 5), though these expressions are no different than ones encountered with *CNP* mRNA. Unfortunately, unlike mRNAs located in neuronal dendrites which are readily detected in high-magnification ISH images, we have not been able to show that *MSAS* mRNAs, other than *MBP* mRNA, reside in MSAS.

## 4. Discussion

We wrote this paper to highlight our concern that studies of myelin-forming oligodendrocytes focus far more on what the cells are doing and far less on where they are doing them. Although we are gaining knowledge of events occurring in myelinating oligodendrocytes per se, we need to put more effort to identify events that locate to myelinating internodes and to characterize the timing of those events during myelin formation and maturation.

Myelin-forming oligodendrocytes extend processes that wrap from one to sixty segments of large caliber axons with myelin sheaths of dimensions sensitive to axons’ calibers and lengths [68,69]. Morphologically, internodes are tethered to the soma by processes structured with a rim of cytoplasm surrounding compact myelin (named MSAS throughout this paper). Each cytoplasmic rim consists of an interconnected outer tongue process, two paranodal loops and an inner tongue process. While many proteins and lipids used to elaborate, maintain and modify compact myelin and surrounding specializations are synthesized in the oligodendrocyte somata, packaged into vesicles and membrane-less particles and transported out to internodes, distinct populations of proteins and lipids are synthesized in each MSAS using mRNAs, proteins and RNAs available to carry out translation (proteins) and metabolic enzymes (lipids). Synthetic activities in each MSAS, mostly along outer tongue processes, furnish proteins that are incorporated into myelin sheaths and responsible for growing and maintaining the internode. To identify the proteins synthesized and functioning in MSAS, one has available a plethora of techniques developed and used to identify and characterize proteins synthesized in neuronal dendrites, axons and in processes of many other cell types [1,70]. A reasonable starting point for these efforts is to compare various screening approaches to identify *MSAS* mRNAs.

With this goal in mind, we developed a screen based on the finding that *MBP* and *MOBP* mRNAs located in MSAS are more efficiently trapped in lipid-rich myelin vesicles than oligodendrocyte somata-residing *PLP1* mRNA [45,71]. Trapped mRNAs separate from others, excepting some which reside in processes of other cells which are close to compact myelin, during ultracentrifugation. Finding that two putative *MSAS* mRNAs, *ENPP2* and *NDRG1*, were not trapped in myelin prepared from tissue used in this study (Figure 1), we considered the possibility that different populations of *MSAS* mRNAs might be trapped during homogenization in solutions of different tonicities. Clearly, using myelin prepared from tissues homogenized in solutions varying tonicity for screens will likely augment the numbers of different *MSAS* mRNAs compared to screens using myelin prepared from tissues homogenized in solutions of single tonicity (next section).

### 4.1. Effectiveness of the Subcellular Fractionation Approach in Identifying MSAS mRNAs

For our initial studies, we used Northern blots to show that *MSAS* mRNAs identified in our screens (Table 1) were enriched in myelin relative to low-speed supernatants used for the fractionations [17,18]. We noted that some *MSAS* mRNAs, e.g., *FHC1* mRNA [18], were larger and/or had larger isoforms than mRNAs in low-speed supernatants, indicating high molecular weight splice variants, as noted for transported mRNAs in other cells [72], may contain sequences used to target, stabilize and/or encode alternatively-spliced variants. Whereas RT-qPCR measures enrichments more precisely than Northern blots, it does not detect high molecular weight splice variants. Future studies should include approaches that identify full-length sequences of *MSAS* mRNAs (Section 4.4.2).

RT-qPCR experiments clearly show that *MSAS* (*MBP* and *MOBP-81A*) mRNAs are more highly enriched (~five-fold and ~twenty-five-fold, respectively) in myelin over oligodendrocyte somata mRNAs (*PLP1*, *CNP* and *MOBP-81B*), and in neuronal (*NEFL*) and astrocyte (*GFAP*) mRNAs (Figure 1). Thus, except for cases in which the mRNAs are expressed at high levels by neurons, astrocytes, microglia and/or ependymal cells, M/P ratio measurements obtained with RT-qPCR will provide a simple and straightforward way to find out if mRNAs of interest reside in MSAS. For example, knowing that *LPAR1* mRNA is enriched in myelin, one could readily design primers and use RT-qPCR to find out if other *LPAR* mRNAs are enriched in myelin as well.

Of thirteen putative *MSAS* mRNAs examined, we showed, based on M/P ratios, that five (*LPAR1*, *TRP53INP2*, *TRAK2*, *TPPP* and *SH3GL3*) reside in MSAS (Figure 1). For *TRP53INP2*, *TRAK2*, and *TPPP* mRNAs, expression by neurons at high levels (Figure 2 and Figure 4) likely yields underestimates of actual levels of these mRNAs in MSAS. Reasons why mRNAs identified in our screens may reside in MSAS despite low enrichment values include: (1) high expression by neurons (*KIF1A*, *MAPK8IP1* and *DYNC1LI2* mRNAs) or ependymal cells (*APOD* mRNA); (2) expression levels which are too low to determine with the RT-qPCR protocol used (*DYNC1LI2* and *PADI2* mRNAs); or (3) exclusion from myelin (*ENPP2* and *NDRG1* mRNAs) resulting from the use of hypertonic sucrose for homogenization. As ISH studies clearly show that *CAR2* [15], *CRYAB* [16], *MATP* [73] and *KIF1A* [17] mRNAs reside in MSAS, either in cultured oligodendrocytes and/or tissues, ISH studies could be used to find out whether or not mRNAs of interest reside in MSAS (Section 4.4.1).

Knowing that presence in neurons lowers enrichments of some *MSAS* mRNAs, using ‘neuron-poor’ white matter—e.g., optic nerve, corpus callosum or spinal cord white matter—as starting material may help reduce influences of neuronal mRNAs, and, consequently, improve identification of *MSAS* mRNAs expressed by neurons. However, based on our experience of using spinal cord white matter in some studies (unpublished observations), we are concerned that using by white matter-rich starting material, the large amounts of myelin accumulating at the 0.32 M/0.85M sucrose interface will trap high levels of ‘non-*MSAS*’ mRNAs than starting tissues with lower myelin content. Increasing dilutions and/or including hypo-osmotic shock may help reduce levels of ‘non-*MSAS*’ mRNAs.

Because we found that *MBP* and *MOBP-81A* mRNAs were trapped at nearly equal levels in myelin prepared from tissues homogenized hypertonic and isosmotic sucrose [44], we conducted these studies using only myelin prepared from tissues homogenized in hypertonic sucrose. We felt the hypertonicity might increase the rate of myelin vesiculation. Finding that *NDRG1* and *ENPP2* mRNAs were not trapped in myelin prepared from hypertonic sucrose, we recommend using both hypertonic and isosmotic sucrose in future studies.

Because all screens were conducted with tissues from rodents at stages past peak (~10 days [74]), myelination as starting material [18,20,44], and because we find that some *MSAS* mRNAs are expressed better by NFOs and other *MSAS* mRNAs are expressed better by MOs (Figure 2), we recommend using tissues from younger, less-myelinated rats in future screens. Furthermore, as myelin from members of all gnathostome classes floats on 0.85 M sucrose [75], we recommend including ‘non-mammalian’ species in future studies. Information on *MSAS* mRNAs from species representing different gnathostome classes will provide an evolutionary perspective as to when mRNAs were recruited for transport to and translation in MSAS. To this end, in looking for *MSAS* mRNAs in spiny dogfish, *Squalus acanthias*, we found that neither *MBP* nor *MOBP* mRNAs were enriched in myelin [76], with results suggesting that neither resided in elasmobranch MSAS, a result suggesting myelination does not require MBP synthesis in MSAS. We also learned that the reason why *MOBP* mRNAs were not purified in elasmobranch myelin was because the MOBP gene family only exists in mammals [76].

In summary, animal species, tissues, development and homogenization conditions are variables worth including when investigators decide to initiate screens to identify *MSAS* mRNAs.

### 4.2. An Independent Approach to Identify MSAS mRNAs

Autoradiographic studies which we used to locate sites of phospholipid and protein synthesis in myelinating Schwann cells show a clear spatial separation of perinuclear and process-associated sites [77,78,79,80]. Similar spatial separation is visible for *MBP* mRNA in ISH images from AMBA and AMSCA (Figure 3 and Figure 5). Based on these images, we realized that by using cryostat sections of fresh tissues stained with DAPI or Hoechst, one could distinguish oligodendrocyte somata nuclei and surrounding perinuclear regions from distant perinuclei-free white matter. Although the peri-nuclei-free white matter likely contains cells and their processes, it is hopefully enriched in *MSAS* mRNAs. If so, separately harvesting nuclei-rich and nuclei-poor regions of cryostat sections with laser capture microdissection [81,82], extracting RNA from pooled samples, preparing cDNA conducting RT-qPCR, should be used to find out if nuclei-free regions are enriched, and by how much, in *MBP* mRNA over *PLP1* mRNA. Enrichments on the order found in subcellular fractions would show that the material could be used for screening. One could increase enrichment further by preparing myelin vesicles from nuclei-free preparations as well. This approach, different from subcellular fractionation (Section 4.1) could be used as an alternative to identify *MSAS* mRNAs. Like subcellular fractionation screens, experiments using tissue samples prepared by laser capture could be applied to different CNS regions, different stages of myelination and to tissues from different species.

### 4.3. Roles Which Identified MSAS mRNAs and Encoded Proteins Might Play in Internodal Myelination

Many proteins used, e.g., for drosophila development [83], neuronal dendritic [84] and axon growth and function [85], astrocyte function at synapses and along blood vessels [86] and radial glial function [87], are synthesized from mRNAs transported to and translated in intracellular sites distant from perinuclear regions of cell somata. Presumably, many proteins participating in myelination are synthesized at least to some extent in *MSAS*. However, until studies which identify them are carried out and the relative contributions of local synthesis versus somatic translation and transport are determined, we are left with the small number of *MSAS* mRNAs identified in our SSH screens and by ISH.

To identify roles which proteins encoded by these few mRNAs might play in myelination, we must place them in a framework of processes believed to occur in MSAS. Myelin sheath formation in MSAS involved major dense line cytoplasmic and interperiod line extracellular leaflet compactions. The former requires MBP [42,88,89], an evolutionarily novel protein [90,91], synthesized in all vertebrate MSAS, except possibly elasmobranch MSAS [92]. Because MBP undergoes post-translational modifications, including deamidation, deimination, N-terminal acylation and phosphorylation [42,88,93], different post-translationally modified MBP variants participate in cytoplasmic membrane compaction [42,88]. As deimination by PADI2 reduces positive charge and influence compaction [35,94,95], and because its mRNA most likely resides in MSAS [18], regulation MBP deimination may include local synthesis of PADI2. To find out if mRNAs encoding other enzymes known to post-translationally modify MBP are similarly regulated, one could use RT-qPCR to determine M/P ratios and ISH approaches to visualize mRNAs in MSAS as a means to see if mRNAs to these enzymes also reside in MSAS.

MBP-regulated compaction involves interactions with lipids at the cytoplasmic surface of oligodendrocyte plasma membrane, among which are polyphosphoinositides [88,96]. Although a great deal of attention has focused on understand how MBP regulates compaction [89,97,98], little has been done to identify the pathways interacting polyphosphoinositides use to get to where locations needed for interactions with MBP. The difficulty of understanding how polyphosphoinositides move among intracellular membranes and the inner surface of the plasma membrane is widespread [99]. In this regard, it would be of interest to know where in myelinated tissues kinases and phosphatases involved in polyphosphoinositide metabolism reside and if any of these are synthesized from mRNAs residing in MSAS.

For major dense line compaction to occur, transmembrane proteins in the oligodendrocyte plasma membrane must be sequestered [100] and cytoplasmic structures, cytoskeleton and organelles removed. Endocytosis and/or autophagy, both associated with myelination [101,102,103] likely contribute to these sequestrations and removals. Among *MSAS* mRNAs enriched in myelin are *SH3GL3* mRNA, encoding endophilin-A3, a protein participating in endocytosis [104,105], and *TRP53INP2* mRNAs, encoding a protein participating in autophagy [106]. Knowing the mRNAs encoding these proteins reside in MSAS, it may be possible to link them to major dense line compaction.

Another mRNA identified in MSAS encodes DYNC1LI2, a cargo adaptor for dynein-mediated retrograde transport [107,108]. This adaptor may function in loading cargoes removed from regions of compaction and/or cargoes transported to soma to signal events in MSAS.

*MSAS* mRNAs associated with kinesin-mediated anterograde transport include *KIF1A*, *MAPK8IP1* and *TRAK2* [109,110,111,112,113]. Mitochondria are known to move around MSAS, contributing to needs for ATP and lipid synthesis [114,115,116,117]. Additionally, screens have shown that mRNAs involved in replenishing mitochondrial proteins are located in MSAS [10,44], and mitochondria are known to transport these mRNAs [118,119].

LPAR1 and ENPP2 influence myelination [120,121,122], with activation of the former increasing process outgrowth in cultured oligodendrocyte and expression of MBP and *MBP* mRNA in these cells [123]. Knowing that LPAR1 and possibly ENPP2 are synthesized in MSAS, possibly in different locations, based on subcellular fractionation results, could mean that if these locally synthesized proteins were functioning in individual internodes, generation of LPA by ENPP2 and activation of LPAR1 by LPA may influence elaboration of myelin at these sites.

*TPPP* mRNA encodes a protein p25/TPPP enriched in oligodendrocytes [37], intrinsically unstructured [124,125], functioning in microtubule bundling and branching [124,126] and locating to Golgi outposts in cultured cells and tissue [39]. Inhibition of TPPP in myelinating oligodendrocytes in vivo and in vitro alter branching and elaboration of myelin sheaths [39]. Adding knowledge that TPPP is synthesized in MSAS may indicate that local synthesis can contribute to oligodendrocyte process branching, elaboration of myelin internodes and possibly recruitment of Golgi outposts, which may, like in dendrites [127,128], contribute to vesicle trafficking, glycosylation and glycolipid metabolism.

Although the ideas as to how identified *MSAS* mRNAs may contribute to internodal myelination are highly speculative, they will hopefully encourage studies which focus on roles local protein synthesis play in elaboration of myelin at individual internodes.

### 4.4. Future Efforts to Identify MSAS mRNAs

To uncover mechanisms which regulate myelin internode development and plasticity, we recognize the need to identify many RNAs, including alternatively-spliced mRNAs, microRNAs and lncRNAs, transported to and functioning in MSAS. Efforts to identify *MSAS* RNAs and determine roles and roles which encoded proteins play in myelination will both enhance conceptual understanding of myelination and contribute to the broader understanding of how RNA transport and local protein synthesis in neurons and many other cell types [1,2,70,129] contribute to function.

#### 4.4.1. Locating Intracellular Sites Where MSAS mRNAs Reside Using ISH

The distinct and recognizable distributions of *MBP* versus *PLP1* and *CNP* mRNAs is apparent in low resolution light microscopic images of tissue sections [130,131] and cultured cells [132]. Low-resolution on-line AMBA and AMSCA images of P56 mouse brain caudate putamen confirm differences in localizations of *MBP* (MSAS) and *CNP* (somata only) mRNA (Figure 9 in [66]), differences seen in P56 mouse brain and spinal cord (Figure 5). Unfortunately, largely due to the fact that other *MSAS* mRNAs are expressed at far lower levels than *MBP*, *MOBP* and *CNP* mRNAs, consequently, exposures used for AMBA and AMSCA images were too short to detect expression of any putative *MSAS* mRNAs identified in our screens or ones identified by ISH (Figure 5).

Hence, it will be necessary to conduct independent ISH studies to determine whether any of these mRNAs and mRNAs identified in future screens reside in MSAS. Fortunately, remarkable advances in single molecule fluorescent ISH (smFISH), as recently reviewed [133,134], will enable investigators to find out if mRNAs and other RNAs identified in screens reside in MSAS. Furthermore, these techniques, alone or modified with tissue expansion protocols [135,136], should allow investigators to distinguish whether expressions occur in outer tongue processes, inner tongue processes, paranodal loops, and/or axoplasm. Applying high resolution ISH will not only enable investigations to find out if mRNAs/RNAs identified in screen reside in MSAS, but also to find out if mRNAs to proteins located in MSAS (e.g., inner tongue processes such as septin and anillin [137,138], myelin-associated glycoprotein [139,140,141] and CADM4 [142]) have mRNAs in these regions. Furthermore, with availability of spatial transcriptomics approaches [143,144,145,146], it should be possible to use these approaches to directly identify *MSAS* mRNAs in tissue sections.

#### 4.4.2. Differential Gene Expression Analysis

Although clearly mRNAs located in MSAS are selectively enriched in myelin vesicles prepared from tissues homogenized in hypertonic and isosmotic sucrose solutions, mRNAs located in oligodendrocyte soma and in other cellular locations can be trapped to varying degrees in myelin vesicles as well. Among the mRNAs identified in our initial screens was *FTH1* mRNA [18]. Our follow up Northern blot analysis showed that the known low molecular weight isoform was not enriched in myelin. Instead, several yet to be characterized higher molecular weight variants were enriched in myelin and, furthermore, these variants were selectively expressed coordinate with *MBP* mRNA during rat brain development. We also found that some of the other mRNAs identified in our screens that were enriched in myelin were larger than corresponding mRNAs in the starting low-speed supernatant (Gould, unpublished). We believe that it will be important to use next-generation sequencing approaches [147] to determine full-length sequences of putative *MSAS* mRNAs. Furthermore, one should be mindful that not only mRNAs will be captured in myelin, but also miRNAs and lncRNAs and that sequencing approaches which identify these RNAs should be considered.

Furthermore, once larger numbers of mRNAs are identified, it seems attractive to apply weighted gene co-expression network analysis analyses (WGCNA, https://horvath.genetics.ucla.edu/html/CoexpressionNetwork/Rpackages/WGCNA/Tutorials/, accessed on 30 March 2023) to identify *MSAS* mRNAs that are functionally linked [148].

#### 4.4.3. Final Thoughts

The focus of this paper is to generate enthusiasm for identifying and characterizing mRNAs which are transported to and are translated in internodal processes, sites allowing independent control of synthesis and interactions that enable internodes to grow with specific regulation from axons being myelinated and from neighboring internodes. We also encourage a broadening of this approach by considering the studies not only in terms of the elaboration of uniquely compacted portions of myelin internodes, but also to proteins underlying functions located at outer and inner tongue processes and in paranodal loops. Clearly high-resolution ISH should help determine whether mRNAs of interest reside in outer versus inner tongue processes versus paranodal loops and axoplasm (Section 4.4.1).

In spite of wide-ranging efforts and a growing interest in myelin lipids and how myelin sheaths become richer in lipids than other plasma membranes [96,149,150,151,152,153,154,155], the idea that some myelin-destined lipids might be synthesized in MSAS, i.e., close to where they enter myelin has, to our knowledge, not been addressed. We learned that a substantial portion of phospholipids incorporated into peripheral nerve myelin is synthesized in superficial cytoplasmic channels [77,78,156], studies backed up by subcellular fractionation studies showing sphingomyelin synthesis occurs in plasma membrane and not in the endoplasmic reticulum [157,158]. We hope that future efforts will include locating sites of metabolism of polyphosphoinositides (Section 4.3) and other myelin-destined lipids.

## Figures and Tables

**Figure 2 life-13-00945-f002:**
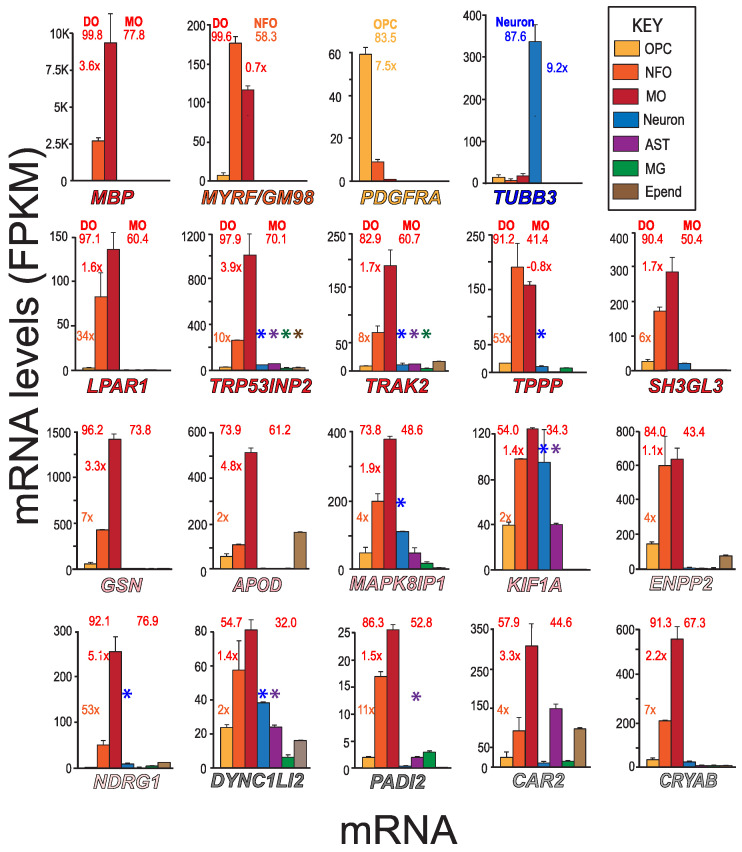
Expressions of different cell-type and *MASA* mRNA—MBRS data. Expression patterns of *MBP*, *MYRF*, *PDGFRA*, *TUBB3* and putative *MSAS* mRNAs are shown for seven neural cell-types acutely isolated from developing mouse brain (MBRS, http://www.brainrnaseq.org, accessed on 30 March 2023). *MSAS* mRNAs are ordered based on overall expression levels (Figure 1). Enrichment values (NFO/OPC, MO/NFO and neuron/(MO + NFO + OPC) ratios) are placed next to NFO, MO and neuron (*TUBB3*) columns, respectively, and percentages of total expressions by differentiated oligodendrocytes, DOs (MOs + NFOs) and by MOs (left and right numbers, respectively, above plots) are given. We add expressions from scRS (http://linnarssonlab.org/cortex/, accessed on 30 March 2023) because neurons used in MBRS do not express some neuronal mRNAs (see below). Positive expressions are shown for neurons, astrocytes, microglia and ependymal cells with colored asterisks (*, *, *, *) above the columns that show mRNA levels.

**Figure 4 life-13-00945-f004:**
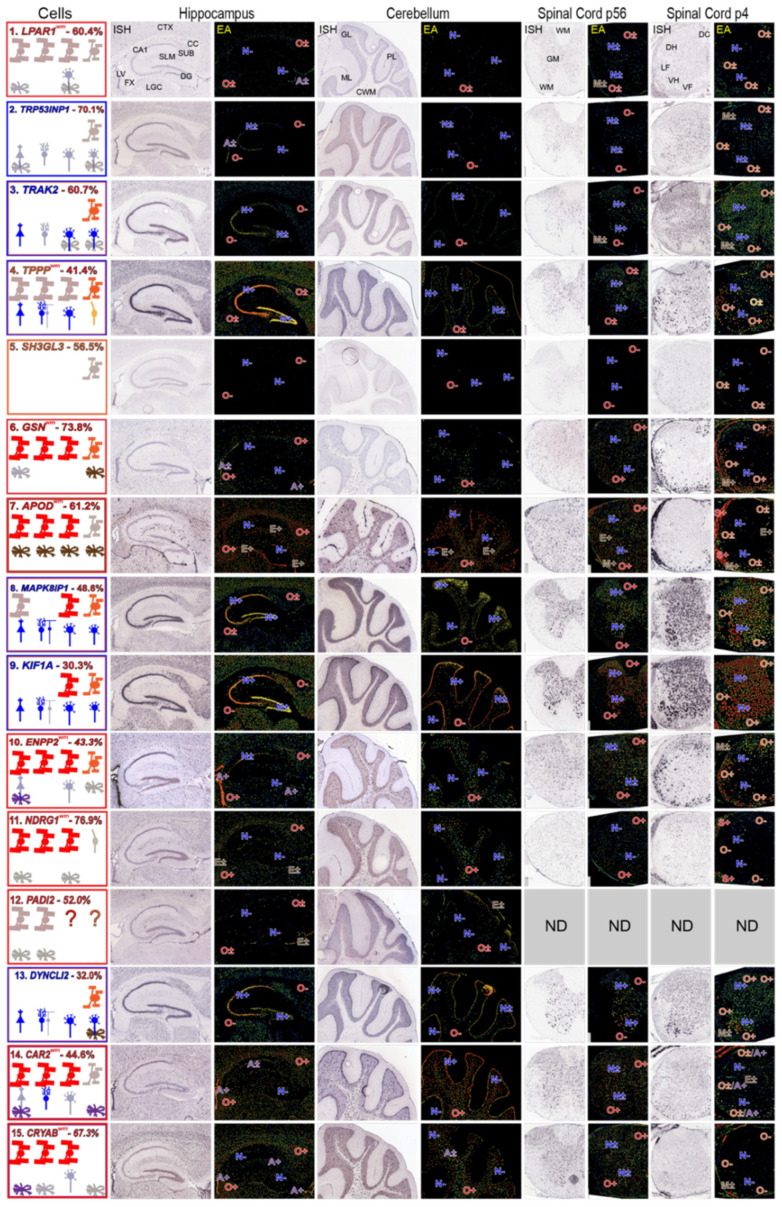
Expression patterns of *MSAS* mRNAs—AMBA and AMSCA images. Images for *MSAS* mRNAs are arranged in rows assembled in the same way as neural cell-specific mRNAs (Figure 3) and according to enrichment levels obtained in subcellular fractionation analyses (Figure 1). Among *MSAS* mRNAs, *SH3GL3* mRNA was not expressed at detectable levels (ne) in P56 brain and spinal cord and *PADI2* mRNA was not included in AMSCA database. Abbreviations identifying brain and spinal cord regions are shown in *LPAR1* mRNA ISH images and cell type expression ratings are recorded in EA images the same as was done for neural cell type expressions. Left panels are boxed in colors representing cell-types expressing the mRNAs, with dominant expressions in the outermost boxes. AMSCA assignments are included following the mRNA name and the percentage of total expression by MOs is included after the mRNA name.

**Figure 5 life-13-00945-f005:**
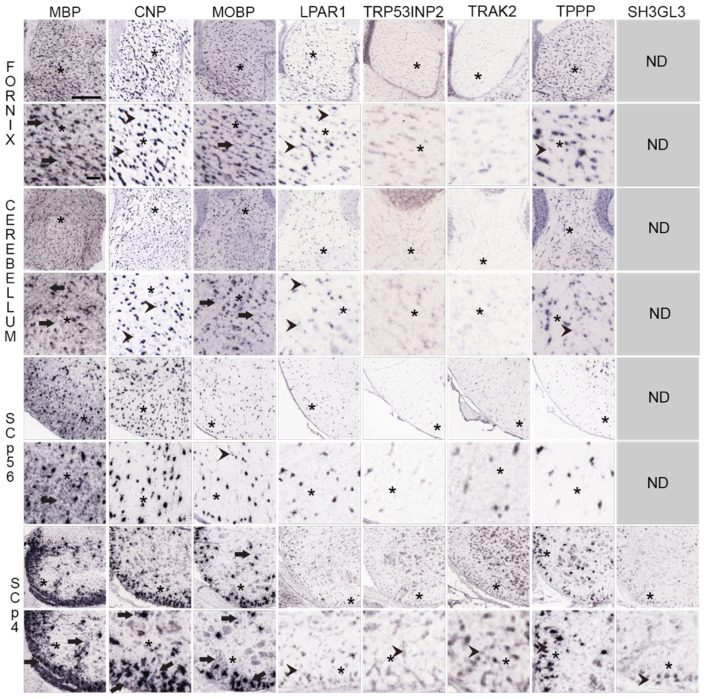
Distribution of *MO* and *MSAS* mRNAs in Mouse Brain and Spinal Cord, high-magnification ISH images. Distribution of *MBP*, *MOBP*, *CNP* and five *MSAS* mRNAs in P56 mouse fornix, cerebellar white matter, P56 and P4 spinal cord white matter. Links to the images are in Appendix A. High-magnification images are placed directly below lower-magnification images and asterisks (*) are used to mark areas in low-magnification images examined at high-magnification. Regions where *MBP* mRNA are expressed in MSAS are shown with arrows as are regions with similar ‘MSAS’ expressions by other mRNAs in high-magnification images. Instances in which we see particle clusters in high-magnification images are marked with arrowheads. As these clusters occur in oligodendrocyte somata-residing images of *CNP*, *PLP1* and *CLDN11*, we consider them to be false-positives. Asterisks (*), represent regions in low-magnification images that are shown in figures below at high-magnification. Scale bar in *MBP* mRNA fornix low-magnification images is 240 μm and for the high-magnification image is 50 μm. Other images are at similar magnifications.

**Table 1 life-13-00945-t001:** Putative *MSAS* mRNAs examined in this study.

#	*MSAS* mRNAs	Gene Symbol	MSAS
1	Apolipoprotein D	*APOD*	
2	Dynein light intermediate chain 2	*DYNC1LI2*	++
3	Ectonucleotide pyrophosphatase/phosphodiesterase 2 ^1^	*ENPP2*	
4	Gelsolin	*GSN*	
5	Kinesin family member 1A	*KIF1A*	++
6	Lysophosphatidic acid receptor 1 ^2^	*LPAR1*	
7	Mitogen-activated protein kinase 8 interacting protein-1 ^3^	*MAPK8IP1*	
8	N-myc downstream regulated 1	*NDRG1*	
9	Peptidylarginine deiminase 2	*PADI2*	++
10	SH3 domain containing GRB2 like 3/endophilin-A3	*SH3GL3*	++
11	Tubulin polymerization promoting protein ^4^	*TPPP*	
12	Trafficking kinesin protein 2 ^5^	*TRAK2*	++
13	Tumor-promoter p53-inducible nuclear protein-2 ^6^	*TRP53INP2*	++
14	Carbonic anhydrase 2	*CAR2*	
15	Crystallin alpha B	*CRYAB*	

^1^ Also known as autotaxin, ^2^ Also known as *EDG2*, ^3^ Also known as *JIP1*, ^4^
*TPPP* was designated as 63OR10 in our publication [18], ^5^
*TRAK2* was designated as 63NA25 in our publication [18], ^6^
*TRP53INP2* was designated as SH3-227 in our publication [18]. Thirteen putative *MSAS* mRNAs identified in our screens [18] plus two others identified by ISH. Six *MSAS* mRNAs were enriched in myelin over levels in starting material used for subcellular fractionation as determined by Northern blot (++). References to mRNAs experimentally linked to oligodendrocyte differentiation and/or myelination are: (1) *APOD* [21,22], (2) *ENPP2* [23,24], (3) *GSN* [25,26], (4) *LPAR1* [27], (5) *NDRG1* [28,29,30,31], (6) *PADI2* [32,33,34,35,36], (7) *TPPP* [37,38,39], (8) *CAR2* [15,40], (9) *CRYAB* [16,41].

**Table 2 life-13-00945-t002:** Amounts of known *MSAS*, oligodendrocyte somata, neuron, astrocyte and *MSAS* mRNAs in myelin vesicles. Comparison of suppression subtractive hybridization (SSH) and mouse myelin transcriptome RNA seq (mmRS) data.

#	mRNA	SSH	%*MBP*	mmRS	%*MBP*	SSH/RS
kMSAS1	* MBP *	44,468 ± 20,315 (10)	100.00	1,265,436 ± 290,298	100.00	1.00
kMSAS2	* MOBP-81A *	1191 ± 225 (5)	2.68	93,503 ± 31,016 *	7.39	0.36
OLS1	* PLP1 *	1096 ± 471 (4)	2.46	88,651 ± 6278	7.01	0.35
OLS2	* MOBP-81B *	36 ± 17 (6)	0.08	--	--	--
OLS3	* CNP *	28 ± 12 (4)	0.06	10,985 ± 597	0.87	0.07
NE1	* NEFL *	8 ± 3 (5)	0.02	5693 ± 255	0.45	0.04
AST1	* GFAP *	11 ± 4 (3)	0.02	5729 ± 296	0.45	0.04
MSAS1	* LPAR1 *	156 ± 102 (10)	0.35	6673 ± 984	0.53	0.66
MSAS2	* NDRG1 *	103 ± 33 (6)	0.23	6334 ± 83	0.50	0.46
MSAS3	* TRAK2 *	41 ± 25 (11)	0.09	19,253 ± 5371	1.52	0.06
MSAS4	* APOD *	33 ± 14 (11)	0.07	12,428 ± 1338	0.98	0.07
MSAS5	* KIF1A *	23 ± 11 (11)	0.05	15,742 ± 2277	1.24	0.04
MSAS6	* TPPP *	19 ± 15 (9)	0.04	9353 ± 2101	0.74	0.05
MSAS7	* SH3GL3 *	17 ± 4 (13)	0.04	4343 ± 347	0.34	0.12
MSAS8	* ENPP2 *	7 ± 2 (11)	0.02	12,016 ± 1371	0.95	0.02
MSAS9	* GSN *	3 ± 2 (8)	0.01	1992 ± 181	0.16	0.06
MSAS10	* MAPK81P1 *	2 ± 2 (8)	0.00	5269 ± 205	0.42	<0.01
MSAS11	* TRP53INP2 *	1 ± 1 (7)	0.00	491 ± 43	0.04	<0.01
MSAS12	* PADI2 *	--	--	760 ± 71	0.06	--
MSAS13	* DYNC1LI2 *	--	--	17,565 ± 4234	1.39	--

* Value is for total *MOBP* mRNA, measured with primers that include all splice variants. Amount ± standard deviation (μg/μl) with numbers of replicates (parentheses) in myelin subfractions were quantified by RT-qPCR with primers listed in Appendix A. Comparisons were made with mRNAs (normalized read count) in P72 mouse brain myelin ± standard deviation (mmRS, mouse brain myelin transcriptome, RNA seq, n = 3, Appendix A [20]). Percentages relative to *MBP* mRNA are given for both data sets. To facilitate comparisons, SSH/mmRS ratios of %*MBP* mRNA were included. Abbreviations: kMSAS, known MSAS; OLS, oligodendrocyte somata; NE, neuron; AST, astrocyte; and putative *MSAS* mRNAs.

## Data Availability

We include a statement that all of the data used is shown or that links were provided to original data sets.

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
