# Peer review of "Identifying mRNAs Residing in Myelinating Oligodendrocyte Processes as a Basis for Understanding Internode Autonomy"

_life, 2023, doi:10.3390/life13040945_

Round 1

Reviewer 1 Report

Many different types of cells localize mRNAs to subcellular compartments where they are translated. Oligodendrocytes are polarized cells that make large amounts of myelin membrane where they contact axons, and mRNA localization and local translation could contribute to myelin synthesis. However, this possibility has not received significant attention in the myelin field. These authors previously applied a subtractive hybridization method to identify mRNAs found in biochemically fractionated myelin, which they reasoned could be mRNAs enriched there by active transport. They identified a handful of mRNAs, which they validated using Northern blotting. In this manuscript they describe using RT-qPCR to test for the presence of the same mRNAs in biochemically fractionated myelin. They then use numerous publicly available data sets to try to evaluate the degree to which their data indicate myelin sheath enrichment.

The manuscript is rather challenging to read and understand. First, the authors create numerous new acronyms, at least one of which is not defined (mmTR, Table 2). These are difficult to keep track of through the manuscript. Second, the manuscript includes lengthy descriptions of numerous gene expression patterns obtained from public databases, which do little to address the point of mRNA localization to myelin. Third, the logic of specific analyses often was not clear or well rationalized.

Despite the numerous gene expression data presented here, the manuscript provides, at best, a minor and incremental advancement. No new localized mRNAs are identified, nor is there any direct validation of their localization to myelin in the brain. The authors take care to provide advice on approaches that other researchers might follow to investigate mRNA localization in myelin, but these are approaches that could have been beneficial to this work. Nevertheless, some readers might find the manuscript useful as a literature review and as an assessment of the myelin mRNA localization field, and as a compendium of gene expression resources.  

Reviewer 2 Report

In this manuscript, Gould and Brady conduct a focused, small-scale screen to identify novel mRNAs that are enriched in myelin sheaths, and thus, that have the potential to be locally translated to contribute to sheath-specific functions. While local translation in neurons has gotten a lot of attention in recent years, our knowledge of local translation in myelin is still in its infancy. Identifying additional mRNAs beyond MBP (the “gold standard”) that are locally translated in myelin sheaths is an important goal for myelin research, with the potential to understand myelin formation, dynamics, and function. Here, Gould and Brady follow up prior work that identified putative myelin-localized mRNAs and validate the myelin enrichment of several using RT-PCR. Overall, this is an interesting manuscript that brings to light an important question in myelin research, contributes new insights, and opens the door to future questions. I have only a few critiques to improve the presentation of the data:

Major point:

(A) Scale bars would be useful in all micrographs. In particular, in Figure 6, without scale bars we have no way of knowing what we are looking at and whether this is sufficient resolution to resolve mRNAs that are strictly in cell somas versus out in myelin sheaths.

(B) It would be useful to include a “caveats” or “limitations of current study” section at the end of the Discussion. Points to mention here include:  (i) Only looked at 13 mRNAs. An unbiased method (e.g. RNAseq, proteomics) has the potential to yield many more hits. (ii) Used 2-3 month old rats, so are more likely to find mRNAs that are locally translated in mature myelin for maintenance. Doing this early postnatally would increase chances of finding mRNAs encoding for proteins required for myelin formation. Etc.

Minor points:

(1) Abstract: The sentence “A reason why others might not be sufficiently enriched is because expression by other cells increase P-values” seems to belong better as part of the discussion than here in the abstract, as it is a detail.

(2) Missing validation experiments. Here are some suggestions, but addressing these caveats in the text would suffice.

a) How sure are they of the purity of “myelin vesicles”? Perhaps a Western blot for myelin-specific proteins would help determine how enriched for myelin proteins these vesicles are compared to total brain lysate and/or the non-myelin pellet fraction.

b) RNA quality control measurements would be useful.

(3) The expression data in Figures 3 and 4 are so zoomed-out that they only really show that there are lots of oligodendrocytes in white matter tracts in the brain. They do not have the resolution to see whether any of the mRNAs that enrich to myelin fractions are actually found in myelin in vivo—that would require zooming much closer in to be able to resolve single oligodendrocyte cell bodies from sheaths. This should be made clear as a caveat of the interpretation here. Thus, I disagree with this sentence: “We believe that the higher white matter expression is a marker of expression in MSAS.” Higher white matter expression just reflects that white matter is highly enriched in oligodendrocytes relative to grey matter, and these mRNAs are highly expressed by oligodendrocytes. Also, minor terminology, but corpus callosum and fornix are not parts of the hippocampus.

Reviewer 3 Report

Dear authors,

The study shows that the expression levels and pattern of myelin enriched mRNA in oligodendrocytes (Myelin sheath assembly sites) by RT-qPCR analysis, RNA-seq analysis, and ISH data. Since authors DYNC1LI2, KIF1A, PADI1, SH3GL3, TRAK2, TRP53INP2 were isolated from pellet and identified as MSAS proteins and discuss well about it, manuscript will be acceptable and suitable in Life without revision.

Author Response

As you accept the manuscript as written, we have nothing further to add, 

Thanks for your effort

Bob Gould